# Interventions for Improving Long COVID-19 Symptomatology: A Systematic Review

**DOI:** 10.3390/v14091863

**Published:** 2022-08-24

**Authors:** Nicola Veronese, Roberta Bonica, Sergio Cotugno, Ottavia Tulone, Michele Camporeale, Lee Smith, Mike Trott, Olivier Bruyere, Luigi Mirarchi, Giuseppina Rizzo, Davide Fiore Bavaro, Mario Barbagallo, Ligia J. Dominguez, Claudia Marotta, Andrea Silenzi, Emanuele Nicastri, Annalisa Saracino, Francesco Di Gennaro

**Affiliations:** 1Department of Internal Medicine and Geriatrics, University of Palermo Geriatric Unit, 90121 Palermo, Italy; 2Department of Biomedical Sciences and Human Oncology, Clinic of Infectious Diseases, University of Bari “Aldo Moro”, 70121 Bari, Italy; 3Centre for Health Performance and Wellbeing, Anglia Ruskin University, Cambridge CB1 1PT, UK; 4Centre for Public Health, Queen’s University Belfast, Belfast BT7 1NN, UK; 5WHO Collaborating Centre for Public Health Aspects of Musculoskeletal Health and Aging, Division of Public Health, Epidemiology and Health Economics, University of Liège, 4000 Liège, Belgium; 6Faculty of Medicine and Surgery, University of Enna “Kore”, 94100 Enna, Italy; 7General Directorate of Health Prevention, Ministry of Health, 00144 Rome, Italy; 8National Institute for Infectious Diseases, Lazzaro Spallanzani, IRCCS, Via Portuense, 292, 00149 Rome, Italy

**Keywords:** long COVID-19, COVID-19, therapy, interventions, SARS-CoV-2

## Abstract

Introduction: Although the understanding of several aspects of long COVID-19 syndrome is increasing, there is limited literature regarding the treatment of these signs and symptoms. The aim of our systematic review was to understand which therapies have proved effective against the symptoms of long COVID-19. Methods: A systematic search for randomized controlled or clinical trials in several databases was conducted through 15 May 2022. Specific inclusion criteria included: (1) intervention studies, either randomized controlled (RCTs) or clinical trials; (2) diagnosis of long COVID-19, according to the World Health Organization criteria; (3) presence of long COVID-19 for at least 12 weeks after SARS-CoV-2 infection. Results: We initially found 1638 articles to screen. After removing 1602 works based on their title/abstract, we considered 35 full texts, and among them, two intervention studies were finally included. The first RCT focused on the greater improvement of treatment combining olfactory rehabilitation with oral supplementation with Palmitoylethanolamide and Luteolin in patients with olfactory dysfunction after COVID-19. The second study evaluated the positive impact of aromatherapy vs. standard care in adult females affected by fatigue. Conclusion: Our systematic review found only two intervention studies focused on patients affected by long COVID-19. More intervention studies are needed to investigate potentially positive interventions for long COVID-19 symptoms.

## 1. Introduction

Coronavirus disease 19 (COVID-19) is a condition that has been putting a strain on health systems around the world for more than two years and has disrupted everyday lives. Parallel to the battle to stem the illness in its most acute phases, the prospect of a new challenge is growing: long COVID-19 [1]. In this regard, some recent systematic reviews and meta-analyses have indicated that most people who contract COVID-19 recover completely, but at least 10–20% of patients experience a range of medium- and long-term effects after recovering from the acute stage of the disease [2]. These medium- and long-term effects are collectively defined as post-COVID-19 syndrome or “long COVID-19”. In the WHO definition, this condition affects individuals with a history of SARS-CoV-2 infection with signs or symptoms occurring 3 months after the COVID-19 infection and lasting for at least 2 months [3].

Interestingly, similar post-viral syndromes have already been observed in conditions caused by herpesviruses, such as cytomegalovirus and Epstein–Barr [4], as well as other coronaviruses such as Middle East respiratory syndrome (MERS) and severe acute respiratory syndrome (SARS) [5]. There is currently limited knowledge on the underlying pathology and epidemiology of long COVID-19; however, studies available are constantly increasing. Long COVID-19 represents a global problem and has the potential to inflict anyone who has developed the SARS-CoV-2 infection regardless of the severity of the acute illness [6,7], even children and adolescents [8].

Several epidemiological studies have shown that the most common symptoms of long COVID-19 are fatigue and dyspnea [9], but other less typical symptoms include myalgia, headache, chest and joint pains, cough, smell and taste dysfunctions, insomnia, wheezing, rhinorrhea, cognitive and mental disorders, and cardiac and gastrointestinal issues. Importantly, these symptoms may persist more than six months [10].

Although many aspects of long COVID-19 syndrome’s understanding are increasing, there is limited literature regarding the treatment. Given this background, the aim of our systematic review is to understand which therapies have proved effective against the symptoms of long COVID-19, which appears to be the chronic manifestation of a disease with complex repercussions [11].

## 2. Materials and Methods

### 2.1. Protocol Registration

This study was conducted following the recommendations in the Cochrane Handbook for Systematic Literature Reviews to conduct the screening and selection of studies [12]. The original protocol was registered in PROSPERO (CRD42022335907). This systematic review and meta-analysis was reported following the Preferred Reporting Items for Systematic Reviews and Meta-Analyses (PRISMA) guidelines [13].

### 2.2. Research Question

The research question for this systematic review is: What are the interventions that can improve long-COVID-19 signs and symptoms? To guide the identification of adequate keywords to build search strategies to search bibliographic databases, the research question was framed into the PICOS (Participants, Intervention, Comparison, Outcome, Study design) format.

The research question formulated into PICO(S) format was, as follows: (P) laboratory confirmed and/or clinically diagnosed COVID-19: long-COVID-19 was defined as the presence of signs and/or symptoms after three months and lasting at least two months and that cannot be explained by other medical conditions, in agreement with the indications of the World Health Organization [WHO] [14]; (I): any pharmacological or non-pharmacological; (C) non active, i.e., placebo or standard/usual care; (O) severity of signs and symptoms of long-COVID-19; (S) randomized controlled (RCTs) or clinical trials (CCTs).

### 2.3. Information Sources and Search Strategies

We searched PubMed, Embase, Scopus, and Web of Science through 15 May 2022, for articles written in English. The search for individual studies in these bibliographic databases was supplemented by manual searches of reference lists of included relevant systematic reviews already published regarding this topic.

Considering the main PICO elements, we built the following search strategy for Medline: (((“COVID-19” [All Fields] OR “Novel Coronavirus-Infected Pneumonia” [All Fields] OR “2019 novel coronavirus” [All Fields] OR “2019-nCoV” [All Fields] OR “SARS-CoV-2” [All Fields]) AND (“lingering symptoms” [All Fields] OR “persistent symptoms” [All Fields] OR “long-term symptoms” [All Fields] OR “long-term COVID” [All Fields] OR “long-term” [All Fields] OR “long-term” [All Fields] OR “long” [All Fields])) OR (“long COVID” [All Fields] OR “long-COVID” [All fields] OR “PACS” [All fields] OR “post acute COVID syndrome” {All fields])) AND (clinicaltrial [Filter] OR randomizedcontrolledtrial [Filter])”. Then, we adapted the search strategy and the syntax for the other databases. The management of the references possibly eligible was performed using the Rayyan website (https://www.rayyan.ai/) accessed on 15 May 2022.

### 2.4. Eligibility Criteria

Specific inclusion criteria included the following: (1) intervention studies, either as RCTs or CCTs; (2) diagnosis of long COVID-19, according to the WHO criteria; (3) presence of long COVID-19 for at least two months [14]. Studies with no WHO definition of long COVID-19 or with an unclear follow-up, observational studies, and head-to-head trials were excluded.

### 2.5. Study Selection

We followed the recommendations reported in the Cochrane Handbook for Systematic Reviews to select studies that were finally included in this review [12]. The selection of the studies was independently carried out by four authors (OT, RB, SC, MC) in two pairs, with consensus meetings to discuss the studies for which divergent selection decision was made. A third senior member (NV, FDG) of the review team was involved when necessary. The study selection process involved first, selection based on title and/or abstracts, and then the selection of studies retrieved from this first step based on the full-text manuscripts.

### 2.6. Data Extraction

Two authors (SC, MC) extracted into a standardized Microsoft Excel sheet the following information (double checked by one senior author, NV): data regarding the identification of the manuscript (e.g., first author name and affiliation, year of publication, journal name, title of the manuscript), characteristics of the population considered (e.g., sample size, mean age, location, gender), setting (e.g., hospital, intensive care unit, hospital), long COVID-19 sign or symptom investigated, method of COVID-19 diagnosis, hospitalized (yes/no/not reported), follow-up (in months), type of intervention. We planned, as declared in the protocol, to extract data for a meta-analysis, but since only two RCTs were available, we reported the data descriptively.

### 2.7. Risk of Bias Assessment

Two authors (OT, RB) independently appraised the risk of bias of the included studies, and one senior (NV) checked these data using the Cochrane Collaboration’s tool through seven aspects of potential biases [12]: random sequence generation, allocation concealment, blinding of participants and personnel, blinding of outcome assessment, incomplete outcome data, selective reporting, and other bias (including early termination and absence of prospective sample size calculation). The risk of bias was then categorized as high, low, or unclear.

### 2.8. Data Synthesis

Since a limited number of studies was available, it was decided to descriptively report the findings.

## 3. Results

### 3.1. Literature Search

As shown in Figure 1, we initially found 1638 articles to screen. After removing 1602 articles based on their title/abstract, we considered 35 full texts and, among them, two intervention studies were finally included [15,16].

### 3.2. Descriptive Characteristics and Risk of Bias

Table 1 shows the main descriptive characteristics of the two RCTs included [15,16], whilst no CCT was available. One study was carried out in Europe and one in North America. In both studies, long COVID-19 was defined according to the criteria proposed by the WHO. These two studies considered mainly females and adults, without including children or older people. As shown in Appendix A, in one RCT [15], we observed a potential high risk of bias for the random sequence generation and unclear regarding allocation since it was not declared, whilst the other work demonstrated low risk of bias [16].

### 3.3. Main Findings

In 1 RCT [16] including 12 participants (followed up for 1 month), in patients with anosmia/hyposmia persisting ≥ 90 days after negative COVID-19 nasopharyngeal swab, the use of an olfactory training/stimulation through Sniffin Sticks, administered twice every day (10-min session) for 30 days plus daily treatment with PEA/Luteolin oral supplement had a greater improvement in the Sniffin score (which measures olfactory sensitivity) than controls who received standard care (mean change in Sniffin score = 2 for controls and 4 for treatment; *p* = 0.01) [16].

The other study included 44 adult females affected by fatigue during long COVID-19 who were followedup for two weeks [15]. In that study, aromatherapy was compared with standard care. Overall, the authors found that people with long COVID-19 affected by fatigue who inhaled the essential oil blend for 2 weeks had significantly lower fatigue scores and that significant results were found on the subscales of global, behavioral, general, and mental fatigue as well as vigor [15].

## 4. Discussion

We conducted a systematic review of interventional studies reporting data that could improve signs and symptoms of long COVID-19, based on the WHO definition [14]. Among all the studies considered, only two were eligible, with a limited sample size and a short follow-up.

The first study focused on olfactory dysfunction [16]. Anosmia and hyposmia are among the most frequent symptoms of both COVID-19 and long COVID-19 and have helped to characterize at the level of general debate what was apparently a simple flu-like syndrome [17]. For more than two decades, semi-quantitative assessment has been the most common study method for olfactory stimulation [18]. Among the most widely used instruments is the Sniffin sticks test. D’Ascanio et al. found that this instrument showed utility in the treatment of olfactory dysfunction from long COVID-19 [16]. The authors reported that the use of olfactory training/stimulation through Sniffin sticks, administered twice daily, showed at 30-day follow-up an improvement in score compared with controls in patients with persistent anosmia/hyposmia ≥ 90 days after COVID-19-negative nasopharyngeal swab [16]. Although the pathogenetic mechanisms underlying COVID-19-related anosmia/dysosmia are still unclear [19], some rehabilitation of the olfactory pathway was achieved in this study. Rehabilitation through sensorial stimulation is reported in other articles, also from the perspective of the self-management of symptoms [20]. This is to be considered an important finding in the management of long COVID-19: As some qualitative assessment studies have documented, the residual symptom of olfactory loss has a strong association with psychological distress in the long COVID-19 patient [21]. Further research on a larger scale is required to verify the findings of this study and inform medical policy regarding long-term olfactory long COVID-19 symptoms.

The second study concerns another common symptom of long COVID-19: chronic fatigue. Through an RCT, Hawkins et al. evaluated the effect of aromatherapy on long-COVID-19-related chronic fatigue. It is known that aromatherapy is useful in other conditions such as the treatment of agitation in patients affected by dementia, whilst conflicting results are present for other diseases [22]. Despite the increased risk of bias compared with the previous study (due to limited data on random sequence generation), the finding that the group that inhaled the essential oil blend for two weeks had significantly lower fatigue scores is important both from a general and mental fatigue perspective [15]. However, these observations were made only in females, limiting the generalizability of these findings in men. However, this study is one of the various attempts to manage chronic fatigue syndrome due to COVID-19. Several authors have agreed that possible treatments of this condition must differentiate possible organic and functional causes [23]. Although a clear and unambiguous pathogenesis has not yet been identified, there is agreement on the roles of other viral infections. EBV is thought to be the major culprit, but cases related to SARS and MERS are also reported in the literature. Only now, however, in view of the important and massive social burden of COVID-19, is the management of a specific virus-related form of chronic fatigue being questioned for the first time [24].

Young adult UK citizens with persistent symptoms following COVID-19 may benefit from an online breathing and wellness program conducted at the English National Opera (ENO), according to the recent ENO Breathe trial. However, because those authors did not apply the WHO criteria for long COVID-19 in their trial, this study is not considered to be eligible for our review [25].

One important recent finding is that vaccination against COVID-19 could be useful in preventing long COVID-19 as shown in a recent large study in the United States of America [26]. The systemic antibody responses probably influence the severity of post-COVID-19/long COVID-19 symptoms, as shown in recent literature [27,28,29]. Although the literature is not univocal regarding this topic [27,28,29], one of the potential mechanisms is the protective role of vaccination against cardiovascular diseases that seems to be present also in long term period [29,30,31,32,33,34].

In order to promote new countermeasures (vaccines, drugs, and diagnostics) for long COVID-19 based on public health requirements, governments, agencies, NGOs, and industry should continue to exchange information, diagnostic tools, and data. This synergy, in the COVID-19 pandemic as in other epidemics [35], could be crucial in controlling long COVID-19 symptomatology.

The findings of our systematic review must be interpreted within its limitations. First, there is a paucity of studies currently available in the published literature. The need to introduce intervention studies for patients with long COVID-19 into the clinical and scientific landscape now arises as imperative: the identification of appropriate symptom management and proper follow-up in the short and long term can thus help to assess possible patient outcomes. In this regard, we found many protocols that will add possible novel treatments for long COVID-19 that in the next future can contribute to new scientific evidence [36].

## 5. Conclusions

In conclusion, our systematic review found only two intervention studies that can help patients affected by long COVID-19. The varied clinical trajectory of long COVID-19, as well as its unclear pathogenesis, raises an additional question, i.e., “Who is the referral specialist for long COVID-19?” To date, management has been left to chance or mere continuity within acute care inpatient units. This might suggest the need to identify the correct characteristics of a multidisciplinary team, which would allow, on the one hand, more efficient care of individual patients, and on the other hand, a more in-depth and timely study of the pathology.

## Figures and Tables

**Figure 1 viruses-14-01863-f001:**
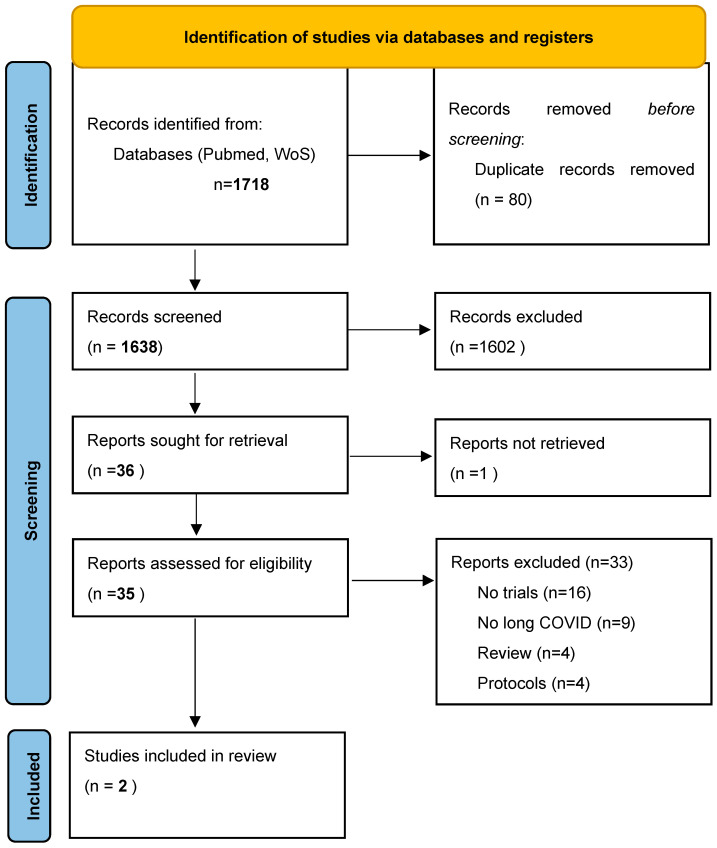
PRISMA flow chart.

**Table 1 viruses-14-01863-t001:** Descriptive characteristics and main findings for the studies included.

Author	Year	Country	Sample Size	Definition of Long COVID-19	Sign or Symptom Investigated	Methods for COVID19 Diagnosis	Age	% Females	Follow-Up (Months)	Type of Intervention	Outcome	Descriptive Results
D’Ascanio	2021	Italy	12	A condition that occurs in individuals with a history of probable or confirmed SARS-CoV-2 infection, usually 3 months from the onset of COVID-19 with symptoms that last for at least 2 months and cannot be explained by an alternative diagnosis	anosmia/hyposmia persisting ≥ 90 days after negative COVID-19 nasopharyngeal swab	nasopharyngeal swab	42.2 (14.1)	67%	1	olfactory training/stimulation through Sniffin Sticks, administered twice every day (10-min session) for 30 days plus daily treatment with PEA/Luteolin oral supplement	Sniffin scores	Patients taking supplement had greater improvement in Sniffin score than controls (mean change in Sniffin score = 2 for CG and 4 for TG; KW: *p* = 0.01)
Hawkins	2022	US	44	fatigue not prior to COVID-19 infection and recovered at least 5 months before	fatigue	nasopharyngeal swab	19–49	100%	0.5	aromatherapy	Multidimensional Fatigue Symptom Inventory, Short Form	Individuals who inhaled the essential oil blend for 2 weeks had significantly lower fatigue scores. Significant results were found on the subscales of global, behavioral, general, and mental fatigue, as well as vigor.

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
