# Peer review of "Interventions for Improving Long COVID-19 Symptomatology: A Systematic Review"

_viruses, 2022, doi:10.3390/v14091863_

Round 1
Reviewer 1 Report
The manuscript by Veronese et al. focuses on the possible therapeutic approach for long COVID. There is still a lack of interventions available, but this systematic review highlights how it is important for researcher to find new approaches for the patients.
Only minor english revision required
Author Response
Reviewer 1
The manuscript by Veronese et al. focuses on the possible therapeutic approach for long COVID. There is still a lack of interventions available, but this systematic review highlights how it is important for researcher to find new approaches for the patients.
Only minor english revision required.
R: We would like to sincerely thank the Reviewer for the appreciation of our manuscript. An English native speaker extensively revised the work.
Reviewer 2 Report
The manuscript under review is an interesting review on the therapies and medical interventions for the treatment of long COVID syndrome. The topic is relevant, considering the impact of long COVID on health and health care. The authors found only two eligible articles, so the actual knowledge on this topic is very poor. The methodology is well explained, and the systematic review is well-conducted. I have only a few minor comments for the authors:
- Lines 49-53: please re-write this sentence, it is too long and there are some repetitions.
- When discussing the study conducted by Hawkins et al., I think it could be useful if the authors add a short explanation of what aromatherapy, if it is an evidence-based treatment, and which kind of essential oil has been used in the mentioned study.
- Even if only two papers have been included in the systematic review, I think the authors should improve the reference list and the discussion with more studies that did fit the inclusion criteria. They may be useful, indeed. For example, there is any correlation between long COVID19 and vaccines? Vaccinated people have less probability to manifest long COVID? Which are the potential pathophysiological mechanisms of long COVID? I suggest citing these papers: 10.1136/bmj.n1648 ; 10.3390/vaccines10020308; 10.1016/j.it.2022.02.008; 10.3390/v14081644; 10.1038/s41591-022-01840-0; 10.1038/s41591-022-01909-w
Author Response
Reviewer 2
The manuscript under review is an interesting review on the therapies and medical interventions for the treatment of long COVID syndrome. The topic is relevant, considering the impact of long COVID on health and health care. The authors found only two eligible articles, so the actual knowledge on this topic is very poor. The methodology is well explained, and the systematic review is well-conducted. I have only a few minor comments for the authors:
- Lines 49-53: please re-write this sentence, it is too long and there are some repetitions.
R:
We would like to sincerely thank the Reviewer for the appreciation of our manuscript. An English native speaker extensively revised the work.
- When discussing the study conducted by Hawkins et al., I think it could be useful if the authors add a short explanation of what aromatherapy, if it is an evidence-based treatment, and which kind of essential oil has been used in the mentioned study.
R: We sincerely thank the Reviewer for this comment. We added the following comment for better explaining the role of aromatherapy in other conditions:
“It is known that aromatherapy is useful in other conditions such as the treatment of agitation in patients affected by dementia, whilst conflicting results are present for other diseases.[22]”
- Even if only two papers have been included in the systematic review, I think the authors should improve the reference list and the discussion with more studies that did fit the inclusion criteria. They may be useful, indeed. For example, there is any correlation between long COVID19 and vaccines? Vaccinated people have less probability to manifest long COVID? Which are the potential pathophysiological mechanisms of long COVID? I suggest citing these papers: 10.1136/bmj.n1648 ; 10.3390/vaccines10020308; 10.1016/j.it.2022.02.008; 10.3390/v14081644; 10.1038/s41591-022-01840-0; 10.1038/s41591-022-01909-w
R: We sincerely thank the Reviewer for indicating us these important works that we have now added to our Discussion section, as follows:
One important recent finding is that vaccination against COVID-19 could be useful in preventing long COVID as shown in a recent large study in the United States of America [26]. The systemic antibody responses probably influence the severity of post-COVID/long COVID symptoms, as shown by recent literature. [27-29] Despite the literature is not univocal regarding this topic [27-29], one of the potential mechanisms is the protective role of vaccination against cardiovascular diseases that seems to be present also in long term period [29-32].
Reviewer 3 Report
I thank the Editor for the opportunity to review the manuscript entitled „Interventions For Improving Long Covid Symptomatology: A Systematic Review”.
This paper is elegant and well written. However, It does not response to an important question. Studies suggest that vaccination before infection may provide some protection in the post-acute phase of the disease, therefore comparison of people with SARS-CoV-2 infection who were not previously vaccinated vs people with breakthrough SARS-CoV-2 infection would have been very interesting particularly focusing on treatment attempts. The systemic antibody responses per se influence the severity of post-COVID/long COVID symptoms (see also: Varnai R, et al. Serum Level of Anti-Nucleocapsid, but Not Anti-Spike Antibody, Is Associated with Improvement of Long COVID Symptoms. Vaccines (Basel). 2022; Ziyad Al-Aly et al. Long COVID after breakthrough SARS-CoV-2 infection. Nat Med 2022). Please mention these in Introduction or among Perspectives. Although the concept of this systematic review was excellent, unfortunately the result contributes modestly to the filed.
Minors: just a few grammatical errors were found.
Regardless, I support the publication of this paper.
Author Response
Reviewer 3
This paper is elegant and well written. However, It does not response to an important question. Studies suggest that vaccination before infection may provide some protection in the post-acute phase of the disease, therefore comparison of people with SARS-CoV-2 infection who were not previously vaccinated vs people with breakthrough SARS-CoV-2 infection would have been very interesting particularly focusing on treatment attempts. The systemic antibody responses per se influence the severity of post-COVID/long COVID symptoms (see also: Varnai R, et al. Serum Level of Anti-Nucleocapsid, but Not Anti-Spike Antibody, Is Associated with Improvement of Long COVID Symptoms. Vaccines (Basel). 2022; Ziyad Al-Aly et al. Long COVID after breakthrough SARS-CoV-2 infection. Nat Med 2022). Please mention these in Introduction or among Perspectives. Although the concept of this systematic review was excellent, unfortunately the result contributes modestly to the filed.
R: We really appreciate this comment. We have now added this reference in the Discussion section, as follows:
“One important recent finding is that vaccination against COVID-19 could be useful in preventing long COVID as shown in a recent large study in the United States of America [26]. The systemic antibody responses probably influence the severity of post-COVID/long COVID symptoms, as shown by recent literature. [27-29] Despite the literature is not univocal regarding this topic [27-29], one of the potential mechanisms is the protective role of vaccination against cardiovascular diseases that seems to be present also in long term period [29-32].”